# Measuring psychological capital: Revision of the Compound Psychological Capital Scale (CPC-12)

**Ludmila Dudasova**[1]\*, **Jakub Prochazka**[1,2], **Martin Vaculik**[1], **Timo Lorenz**[3]

**1** Department of Psychology, Faculty of Social Studies, Masaryk University, Brno, Czech Republic,
**2** Department of Corporate Economy, Faculty of Economics and Administration, Masaryk University, Brno, Czech Republic, **3** Department of Psychology, Medical School Berlin, Berlin, Germany

\* Lud.dudasova@mail.muni.cz

## Abstract

This article provides information about the psychometric limitations of the original Compound Psychological Capital Scale (CPC-12) and suggests a revised version CPC-12R, a free-to-use measure of Psychological Capital. The investigation consisted of three studies: two of these identified psychometric limitations of the original scale, and the third presented the revised version of the scale. The first study did not confirm the hypothesized four-factor structure of the CPC-12 on a sample of Czech teachers ($n = 282$) and found psychometric limitations in the resilience subscale. The second study identified the same problem using secondary analyses of the original data from two samples of German employees ($n = 202$ and $321$ respectively). The third study proposed a revised version of the scale with new items for resilience, and provided support for reliability and factorial validity of the new CPC-12R on a sample of Czech employees ($n = 333$). CPC-12R demonstrated a better fit to the theoretically supported model of Psychological Capital than CPC-12, and further displays adequate psychometric properties to be recommended for application in both research and practice.

## Introduction

The construct of Psychological Capital (PsyCap) [1] draws from positive psychology in general [2] and positive organizational behavior [3] in particular. It broadens the traditional pair of human and social capital as it represents

> '*an individual's positive psychological state of development that is characterized by: (1) having confidence (self-efficacy) to take on and put in the necessary effort to succeed at challenging tasks; (2) making a positive attribution (optimism) about succeeding now and in the future; (3) persevering toward goals and, when necessary, redirecting paths to goals (hope) in order to succeed; and (4) when beset by problems and adversity, sustaining and bouncing back and even beyond (resilience) to attain success.*'

[4 p3].

**Data Availability Statement:** All relevant data are within the manuscript and its Supporting information files.

**Funding:** This study was supported by the project When Close Relationships Matter: A Longitudinal

Study of Psychological Capital Development (GA20-03810S) of the Czech Science Foundation GACR. Full name of the funder: Grantová agentura České republiky Funder website: https://gacr.cz/ This did not play any role in the study design, data collection, analysis etc.

**Competing interests:** The authors have declared that no competing interests exist.

The existence of a higher order core construct, PsyCap, has both conceptual [1] and empirical [5] support. The commonalities among the above mentioned four sub-dimensions of PsyCap–hope, self-efficacy, resilience, and optimism–allow PsyCap to be considered as a core construct [6] that contains these sub-dimensions. Research has shown that each of these sub-domains has conceptual independence and empirically established discriminant validity [7]. The four sub-domains share a sense of control, intentionality, and agentic goal pursuit [8], which are also characteristics of PsyCap.

Since its theoretical definition in 2004 [1], PsyCap has received considerable attention both in research and in practice, and positive implications for human-resource development have been reported repeatedly. In a meta-analysis including 51 independent samples Avey et al. [9] found that PsyCap significantly predicts well-being, desirable employee attitudes (job satisfaction, organizational commitment, psychological well-being), desirable employee behavior (organizational citizenship), and multiple measures of employee performance (self-rated, supervisor-rated, and objective). They also found a statistically significant negative relationship between PsyCap and undesirable employee attitudes (cynicism, turnover intentions) and behavior (deviance). Moreover, although PsyCap predominately focuses on positivity at the individual level, positive associations between collective PsyCap and team performance have also been demonstrated [10, 11].

A holistic approach to PsyCap involves examining its effect across multiple life domains, including work as well as relationships and health [12, 13]. Experimental studies have supported PsyCap development and change through relatively short training interventions [14–17] including web-based interventions [5]. Given the evidence that PsyCap represents a construct with the potential to positively influence both individuals and whole organizations, the need for valid and accessible diagnostic tools to measure PsyCap for different aspects of life and cultural contexts is evident.

## Measuring Psychological Capital

The Psychological Capital Questionnaire (PCQ) [6] is recognized as the standard scale measuring PsyCap [18]. It was developed as a compound measure consisting of modified items from pre-existing published scales for hope (State Hope Scale) [19], optimism (Life Orientation Test) [20], resilience (Resilience Scale) [21], and self-efficacy (Role Breadth Self-Efficacy Scale) [22]. However, despite endorsement of the PCQ in literature, there is room for improvement, particularly in relation to test-retest reliability and convergent and discriminant validity [18]. Moreover, the questionnaire includes items difficult to use in small organizations (e.g., "I feel confident contributing to discussions about the company´s strategy") or outside the work environment. For use within this scope, items need to be adapted [13]. Another obstacle to widespread use of the questionnaire may be that the method is licensed: consequently, researchers must meet several criteria to receive a special permission before they use it, and practitioners who want to use it for consulting, training, or any similar function, must pay for it, which may restrict PsyCap interventions and its utility in the workplace. Although authors of PCQ provide translations in almost 30 languages, psychometric evaluation of some of these translations is lacking [23]. In validation studies of translated versions, researchers repeatedly found factor structures that did not correspond to the theory of four-factor structure (see, e.g., the three-factor Indian version of Sahoo and Sia, [24]; the five-factor Portuguese version of Rego et al., [25]; the one-factor Chinese version of Cheung et al., [26]). Based on these findings, caution is necessary while interpreting results. It is important to consider whether the cause of discrepancies is rooted in poor adaptations or psychometric deficiencies of PCQ itself.

An alternative measure of PsyCap is the semi-projective Implicit Psychological Capital Questionnaire (I-PCQ) by Harms & Luthans, [27], which estimates the level of PsyCap based on an evaluation of characters in three fictional stories. In a validation study, Harms et al. [28] found support for I-PCQ as a valid predictor of work attitudes and behavior and found moderate convergence between PsyCap measured by implicit PCQ (I-PCQ) and explicit PCQ (PCQ-12). However, the study also reported psychometric shortcomings of the method in terms of inadmissible parameter estimates and encouraged further verification of the psychometric properties of this instrument before it can be recommended for use in practice. Consequently, with a view to expanding research and application opportunities in small organizations as well as in other domains such as sports and education, Lorenz et al. [29] designed and validated a new universal measure for the construct, Compound PsyCap Scale (CPC-12), and provided it with an open access license. The scale consists of 12 items selected from 5 existing scales for hope (State Hope Scale, [19]), optimism (revised German short version of the Life Orientation Test, and Affektive Valenz der Zukunftsorientierung [30, 31]), resilience (German short version of the Resilience Scale [32]), and self-efficacy (German General Self Efficacy Scale and German short version of the Occupational Self-efficacy Scale [33, 34]). While PCQ-12 is composed of three items for self-efficacy, four for hope, three for resilience, and two for optimism [8], the distribution of items measuring each subscale is even in CPC-12 (three for each); hence, the effect of all components on composite PsyCap score is balanced.

Using two independent samples (*n* = 334 and 202 respectively), Lorenz et al. [29] found promising evidence supporting the validity of CPC-12. They showed good fit of the data to the theoretical model (four first-order factors and PsyCap as a second-order factor), high internal consistency of the compound scale of PsyCap, and a high correlation of PsyCap measured by CPC-12 with PsyCap measured by PCQ. Moreover, Lorenz et al. [29] found support for the construct validity of CPC-12, showing that the correlations between various related constructs and psychological capital measured by CPC-12 are in line with existing theory. Some studies also showed that the data collected by CPC-12 fit the theoretical model of PsyCap well (see, e. g., Khajavy et al., [35]). However, the CPC-12 is still a new scale and more robust evidence on its validity is missing, especially if it is to be used cross-culturally. As for factorial validity, neither Lorenz et al. [29] nor other authors (e.g., Pajic et al., [36]) compared the four factor model with alternative models that might also explain the structure of CPC-12, except for the comparison of models with and without the second-order factor [35]. With regards to reliability, Lorenz et al. [29] did not provide coefficients of internal consistency of specific CPC-12 subscales. According to Khajavy et al. [35], the internal consistency of resilience and optimism subscales is slightly lower than α = .70. However, the lower consistency might be explained by the fact that the respondents of their research were not native English speakers.

Our original aim was to adapt and validate a Czech version of CPC-12 to enable measurement of and research on PsyCap in Czech organizations and to provide further evidence about the validity and reliability of CPC-12. However, when validating the Czech CPC-12, we found psychometric limitations that appeared to be connected to the original instrument as well as to the Czech version (see Study 1). We performed a secondary analysis of the data from the original validation study [29] and found the same psychometric limitations (see Study 2). Therefore, our aim has now changed to show the psychometric limitations of the original CPC-12 and suggest a revised version, which deals with the limitations of the original scale.

## Study 1

The aim of the first study was to find support for the hypothesis that the data collected by Czech translation of CPC-12 fit the theoretical model of PsyCap (i.e., four first-order factors: hope, self-efficacy, optimism, and resilience; one second-order factor: PsyCap).

### Materials and methods

**Psychological Capital.**  PsyCap was evaluated using the Czech translation of Compound Psychological Capital Scale CPC-12 [29]. The scale consists of twelve self-evaluating statements rated on a 6-point Likert scale (ranging from 1 = *strongly disagree* to 6 = *strongly agree*). The translation process followed the guidelines for translating and adapting psychological instruments [37]. Three persons fluent in both English and Czech translated the instrument parallel. The inconsistencies between the independent translations were settled by a PsyCap expert. This version of the instrument was further back translated to English by two independent professionals and results were compared by a third party. Finally, two waves of cognitive interviews (*n* = 5) were carried out; the first to test language and meaning of items and instructions for administration, and the second to check the effects of changes made based on the first pilot study.

**Participants and procedure.**  The sample in Study 1 consisted of 282 teachers. Average age of the participants was 44.09 years, (*SD* = 11.32), and average work experience was 18.50 years (*SD* = 11.39). A majority of the sample were women (84%), and 15% were men. One participant did not specify binary gender. There were 28% primary school teachers, 24% taught at a secondary school, 2% at an apprentice school, 30% at a college, and 16% at a grammar school. Participants were recruited by publishing the link to the survey in several online social media groups, and by emails sent to directors of selected schools (i.e., by convenience sampling). The survey was conducted in Czech. Participation was voluntary with no compensation. Participants were informed that they would give their consent by proceeding past the welcome page of the online survey. We did not seek ethics committee approval in this case as data were gathered anonymously and we did not expect any risks of harm for participants.

### Results

Table 1 presents descriptive statistics for summary scores of all four PsyCap subscales and for the compound scale. The internal consistency of the Resilience subscale was very low. A more detailed inspection of the items showed that the first (r = .355) and the third (r = .238) resilience items had very low item-rest correlations when used as a part of the 12-item compound PsyCap scale.

We conducted confirmatory factor analysis (CFA) using MPLUS 8 [38] with a maximum likelihood robust estimation (MLR) to test the same model as Lorenz et al. [29] tested in the validation study of CPC-12. The model with 12 items, four first-order factors (i.e., hope, self-efficacy, optimism, and resilience), and one second-order factor (i.e., PsyCap) did not converge.

We then tested an alternative model with four correlated factors (i.e., hope, self-efficacy, optimism, and resilience). This alternative model was estimated with good fit ($\chi^2(48)$ = 104.204; confirmatory fit index CFI = .954; Tucker-Lewis index TLI = .937; Root Mean Square Error of Approximation RMSEA = .064; 90% confidence interval CI $_{RMSEA}$ [.047, .081]). However, reliability of the result was dubitable because of the linear dependency of two factors. The correlation between factors self-efficacy and resilience (*r* > .999) indicated that the latent variables self-efficacy and resilience were extremely close to each other. Therefore, the results did not provide support for our hypothesis.

**Table 1. Descriptive statistics for summary scores of CPC-12 subscales and the whole scale.**

|  | *M* | *SD* | Hope | Optimism | Resil. | S-E | α | ω |
|---|---|---|---|---|---|---|---|---|
| Hope | 13.149 | 2.410 |  |  |  |  | .731 | .738 |
| Optimism | 14.227 | 3.093 | .572** |  |  |  | .885 | .891 |
| Resilience | 13.089 | 2.286 | .454** | .388** |  |  | .434 | .467 |
| Self-efficacy | 14.004 | 2.300 | .629** | .488** | .578** |  | .813 | .821 |
| PsyCap | 54.470 | 8.131 | .831** | .803** | .732** | .825** | .902 | 886 |

Note.

**p < .001; resil = resilience; S-E = self-efficacy

To effectively understand the result, we conducted a supplementary analysis in which we tested a model with only three first-order factors (hope, optimism, and a third factor that merged self-efficacy and resilience) and one second-order factor of PsyCap. According to fit indices ($\chi^2$(51) = 105.038; CFI = .956; TLI = .943; RMSEA = .061; 90% $CI_{RMSEA}$ [.045, .078]), the model had satisfactory fit and should be preferred over the four-factor model. Table 2 shows standardized factor loadings. As can be seen from the table, two items from Resilience subscale (items one and three) had low factor loading on common factor.

## Discussion

Lorenz et al. [29] recently proposed a new method of PsyCap measurement, CPC-12, that overcomes some of the shortcomings of the PCQ [18]. The original aim of this study was to translate and validate a Czech version of CPC-12 in order to bring new supporting evidence to establish CPC-12 as a valid measurement tool, and to promote research and PsyCap interventions in Czech workplaces. However, the resilience subscale showed low internal consistency, and its items did not correlate with the rest of the compound PsyCap scale. Moreover, the results of our study suggested that the Czech CPC-12 did not fit the theoretical model of Psy-Cap. A three-factor model of PsyCap was more suitable, as the latent variables of self-efficacy and resilience appeared to be linearly dependent. Although the latent variables were linearly dependent, the correlation between summary scores of resilience and self-efficacy was only moderate. This was probably caused by low reliability of the resilience subscale, and, therefore, the high measurement error. The very high correlation between the latent variables was probably caused by very low factor loadings of two resilience items (one and three). The latent variable resilience explained especially the variance of the second resilience item which had content similar to that of self-efficacy items. To explain our findings, we identified three possible reasons:

1. The findings could be a result of mistaken translation. We might have changed the content of resilience items or formulated the items from the self-efficacy and resilience subscales closer together, wiping out the differences between the correlated factors.

2. The results could reflect an error in PsyCap theory. The constructs of self-efficacy and resilience could be so similar that they should compose a common factor rather than two separate ones.

3. The results could reflect a limitation in the original CPC-12. The authors of the original study did not provide reliability coefficients for particular subscales; and they also did not test alternative models. It is possible that there is an unreliable resilience subscale and sub-optimal data fit even in the original version of CPC-12.

**Table 2. Standardized factor loadings in model with 3 first order factors and 1 second order factor.**

|  | Hope | Optimism | S-E/Resil. | PsyCap |
|---|---|---|---|---|
| Hope1 | .670 |  |  |  |
| Hope2 | .686 |  |  |  |
| Hope3 | .723 |  |  |  |
| Optimism1 |  | .861 |  |  |
| Optimism2 |  | .930 |  |  |
| Optimism3 |  | .773 |  |  |
| Resil1 |  |  | .372 |  |
| Resil2 |  |  | .769 |  |
| Resil3 |  |  | .325 |  |
| S-E1 |  |  | .857 |  |
| S-E2 |  |  | .753 |  |
| S-E3 |  |  | .703 |  |
| Hope |  |  |  | .988 |
| Optimism |  |  |  | .689 |
| S-E/Resil. |  |  |  | .836 |

*Note*: resil = resilience; S-E = self-efficacy

To inspect the first possible cause of our findings, we focused on the quality of translation of Czech items. However, we concluded that the multi-stage translation was performed well, and the content of the Czech items correspond to the content of the original items.

Consequently, we investigated the second explanation. According to theory, self-efficacy and resilience indeed are close constructs. When applied to the workplace, resilience is defined as the "positive psychological capacity to rebound, to 'bounce back' from adversity, uncertainty, conflict, failure, or even positive change, progress and increased responsibility" [3 p702]. Thus, resilience refers to rapidly returning to baseline functioning after exposure to an adverse situation. Self-efficacy refers to "an individual's conviction (or confidence) about his or her abilities to mobilize the motivation, cognitive resources, and courses of action needed to successfully execute a specific task within a given context" [39 p66]. It comprises a sense of control over one's environment and an optimistic belief of being able to successfully alter challenging environmental demands by means of one's own behavior. Hence, individuals with high levels of perceived self-efficacy trust their own abilities in the face of adversity, tend to conceptualize problems as challenges rather than as threats or uncontrollable situations, experience less negative emotional arousal in demanding tasks, think in self-enhancing ways, motivate themselves, and show perseverance when confronted with adverse situations [40, 41]. Consequently, being self-efficacious may be helpful to show resilience in the face of adversity. By activating affective, motivational, and behavioral mechanisms in problematic situations, self-efficacy beliefs promote resilience. In line with this, self-efficacy has sometimes been conceptualized as one component of resilience [42, 43], which is supported by newer empirical research documenting moderate to strong correlations between self-efficacy and resilience [44–47]. Still, they are not the same: while self-efficacy may be present even in the absence of stressors, one cannot be resilient if there is no stressful event [48].

Interestingly, the independence of self-efficacy and resilience in the original PCQ [1] has never been questioned. The distinction of both factors is supported by the fact that the correlations between self-efficacy and resilience factors were only moderate ($r = .40$, $.42$, and $.43$; $p < .05$, [4]). Moreover, several other studies which measured both constructs showed that self-

efficacy and resilience are correlated but different constructs [44, 46]. Therefore, it seems that the four-factor model of PsyCap should be valid, and the problematic fit of our data to the four-factor model is in fact a weakness of the instrument. The theoretical evidence together with empirical evidence lead us to explore the third hypothesized cause of our findings.

Although the CPC-12 [29] has already gained much attention and has been cited repeatedly, we did not find any study that would bring evidence of the four-factor model being better than more parsimonious models, which creates doubts about a three-factor model being a better fit for the data. We also did not find evidence about the high reliability of the resilience subscale. In order to determine whether the low internal consistency and high proximity of latent factors of resilience and self-efficacy is a general problem of CPC-12 or only persists in the Czech translation of the questionnaire, a secondary analysis of data published by Lorenz et al. [29] and a comparison of its results with the data obtained by the Czech version of the questionnaire was performed in Study 2.

## Study 2

Study 2 represents an exploratory analysis of data from the two German samples based on which Lorenz et al. [29] created and validated the original German version of CPC-12.

Although, according to theory and empirical evidence provided by Luthans et al. [4], the structure of psychological capital is four-factor, we hypothesized that there is an overlap between the factors of resilience and self-efficacy in CPC-12 and that the model with four first-order factors does not explain data better than a model with three first-order factors (i.e., with items from self-efficacy and resilience subscales merged in one common factor).

### Materials and methods

For the purpose of the analysis, we used data collected by Lorenz et al. [29] for the CPC-12 (see description of this measure in Study 1).

**Participants and procedure.** The first sample consisted of 321 participants ($M$ = 34.89 years, $SD$ = 12.78), with a slight predominance of women (60%). Of the participants, 76.6% were employees, 13.7% temporary workers, and 8.4% were self-employed. The second sample consisted of 202 participants with an average age of 37.79 years ($SD$ = 13.10), of which more than two-thirds were women (72.3%). The sample consisted of 82.7% employees, 9.4% self-employed workers, and 7.9% temporary workers. Participants were recruited by publishing a link to the survey in several online social media groups, and all participated voluntarily. They were informed that they would give their consent by proceeding past the welcome page of the online survey. No compensation was supplied [29].

### Results

Descriptive statistics for all CPC-12 subscales in both German datasets are shown in Table 3. Similar to the Czech sample, the Resilience subscale suffers from low internal consistency in both German samples.

We performed confirmatory factor analyses (CFA) with MLR estimator using both the datasets published by Lorenz et al. [29] to compare the original model with four first-order factors (i.e., hope, self-efficacy, optimism, and resilience) and one second-order factor (i.e., PsyCap) with alternative models that merged resilience and self-efficacy items into one common first-order factor. According to fit indices (see Table 4), the more parsimonious model with three first-order factors should be preferred over the model with four first-order factors. There is no need to test the difference as the more parsimonious model had lower chi-square value and RMSEA, and higher CFI than the complex model. This result supports our hypothesis.

**Table 3. Descriptive statistics for summary scores of CPC-12 in German datasets.**

| | Dataset 1 (N = 321) | | | | Dataset 2 (N = 202) | | | |
|---|---|---|---|---|---|---|---|---|
| | M | SD | ω | α | M | SD | ω | α |
| Hope | 12.835 | 2.344 | .743 | .717 | 13.198 | 2.264 | .737 | .715 |
| Optimism | 13.745 | 2.076 | .718 | .711 | 15.010 | 2.213 | .783 | .781 |
| Resilience | 13.667 | 1.897 | .483 | .416 | 13.822 | 1.795 | .509 | .427 |
| Self-efficacy | 12.181 | 1.976 | .672 | .668 | 12.401 | 2.023 | .713 | .709 |

When using data from the second German dataset, the model with four first-order factors has slightly better values of Goodness-of-fit indices than the more parsimonious model with three first-order factors (see Table 5). However, the difference in the fit indices was small ($\Delta$CFI < .01, $\Delta$RMSEA < .005), and the difference in CFI is even smaller than the critical value for rejecting the null hypothesis of equivalence [49]. As the fourth factor in the model did not contribute to a significant improvement in the model's fit, the more parsimonious model with three factors should be preferred over the model with four factors which can be considered as a support for our hypothesis.

Although the correlations between resilience and self-efficacy in the German sample (correlation of latent variables: $r$ = .719 for the first German dataset; $r$ = .709 for the second German dataset) were not as large as in the Czech dataset, resilience and self-efficacy subscales were close to each other. As can be seen from Tables 6 and 7, items one and three that measure resilience had rather low factor loading on the resilience factor (see model with 4+1 factors), which is consistent with the results obtained on the Czech sample (see Study 1).

## Discussion

Since its publication in 2016, the Compound PsyCap Scale (CPC-12) has attracted the attention of researchers and has been used in different cultural contexts, e.g., for Syrian refugees [36], Chinese students [50], and Indian mothers [51]. Based on psychometric evaluation, the Czech version failed to present support for factorial validity. Consequently, the aim of this study was to explore the psychometric qualities of the original CPC-12 and compare them with those of the Czech version (described in Study 1).

According to our results, the original CPC-12 had shortcomings similar to those we identified in the Czech version, i.e., the resilience subscale was not reliable, the distinction between self-efficacy and resilience factors did not contribute to a significant improvement in the model's fit and two of the resilience items showed to have rather low factor loadings.

Consequently, we decided to inspect the content validity of resilience and self-efficacy subscales, reconsider the choice of items measuring resilience and self-efficacy, and replace some

**Table 4. Comparison of various models using first German dataset (N = 321).**

| Model | Description | $\chi^2$ | df | p | CFI | TLI | RMSEA [90%CI] | SRMR |
|---|---|---|---|---|---|---|---|---|
| Baseline | Uncorrelated items | 762.382 | 66 | < .001 | < .001 | < .001 | .181[.170, .193] | .244 |
| 1 factor | 12 items on PsyCap | 190.189 | 54 | < .001 | .804 | .761 | .089 [.075, .102] | .069 |
| 3 factors | S-E + resil. merged | 62.322 | 51 | < .001 | .984 | .979 | .026 [.000, .047] | .041 |
| 2nd order 3f | 3 first-order f. + PsyCap | 62.321 | 51 | < .001 | .984 | .979 | .026 [.000, .047] | .041 |
| 2nd order 4f | 4 first-order f. + PsyCap | 62.461 | 50 | < .001 | .981 | .974 | .029 [.000, .049] | .042 |

*Note*: resil = resilience; S-E = self-efficacy; first-order f = first-order factor; 3f = 3 factor; 4f = 4 factor

**Table 5. Comparison of various models using second German dataset (N = 202).**

| Model | Description | $\chi^2$ | df | p | CFI | TLI | RMSEA [90%CI] | SRMR |
|---|---|---|---|---|---|---|---|---|
| Baseline | Uncorrelated items | 625.711 | 66 | < .001 | < .001 | < .001 | .205[.190, .220] | .258 |
| 1 factor | 12 items on PsyCap | 202.29 | 54 | < .001 | .735 | .676 | .117 [.100, .134] | .085 |
| 3 factors | S-E + resil. merged | 80.823 | 51 | < .001 | .947 | .931 | .054 [.030, .075] | .054 |
| 2nd order 3f | 3 first-order f. + PsyCap | 80.823 | 51 | < .001 | .947 | .931 | .054 [.030, .075] | .054 |
| 2nd order 4f | 4 first-order f. + PsyCap | 74.952 | 50 | < .001 | .955 | .941 | .050 [.024, .072] | .052 |

*Note*: resil = resilience; first-order f = first-order factor; 3f = 3 factor; 4f = 4 factor

of the items in order to improve psychometric qualities of the scale. We describe the process and introduce the revised version CPC-12R in study 3.

## Study 3

In this study, we aimed to improve the existing CPC-12 so that the instrument would measure resilience more reliably and would better differentiate between resilience and self-efficacy. As described in Studies 1 and 2, factor analyses of one Czech and two German datasets indicated that the model with four first-order factors did not explain the data better than the model with only three first-order factors owing to a suspected overlap between the subscales measuring self-efficacy and resilience. The problem might be caused by (1) the wording of the first and last resilience items which seem to measure different construct, and (2) the content of second resilience item, and/or content of self-efficacy items which seem to be too close to each other. Therefore, we examined the items intended to measure resilience and self-efficacy and checked if they reflected the definitions of the constructs.

Self-efficacy is defined as *"having confidence to take on and put in the necessary effort to suc-ceed at challenging tasks"* [4 p3]. In CPC-12, it is measured in a specific context of dealing with

**Table 6. Standardized factor loadings in models with three and four first-order factors and one second-order factor (first dataset, N = 321).**

| | 3+1 factors | | | | 4+1 factors | | | | |
|---|---|---|---|---|---|---|---|---|---|
| | Hope | Optimism | S-E/Resil. | PsyCap | Hope | Optimism | Resil. | S-E | PsyCap |
| Hope1 | .586 | | | | .595 | | | | |
| Hope2 | .679 | | | | .676 | | | | |
| Hope3 | .812 | | | | .806 | | | | |
| Optimism1 | | .591 | | | | .588 | | | |
| Optimism2 | | .796 | | | | .801 | | | |
| Optimism3 | | .637 | | | | .634 | | | |
| Resil1 | | | .293 | | | | .333 | | |
| Resil2 | | | .628 | | | | .723 | | |
| Resil3 | | | .410 | | | | .386 | | |
| S-E1 | | | .588 | | | | | .620 | |
| S-E2 | | | .626 | | | | | .637 | |
| S-E3 | | | .618 | | | | | .650 | |
| Hope | | | | .949 | | | | | .836 |
| Optimism | | | | .625 | | | | | .617 |
| Resil. | | | | .756 | | | | | .890 |
| S-E | | | | | | | | | .808 |

*Note*: resil = resilience; S-E = self-efficacy

**Table 7. Standardized factor loadings in models with three and four first-order factors and one second-order factor (second dataset, N = 202).**

| | 3+1 factors | | | | 4+1 factors | | | | |
|---|---|---|---|---|---|---|---|---|---|
| | Hope | Optimism | S-E/Resil. | PsyCap | Hope | Optimism | Resil. | S-E | PsyCap |
| Hope1 | .637 | | | | .659 | | | | |
| Hope2 | .637 | | | | .624 | | | | |
| Hope3 | .795 | | | | .780 | | | | |
| Optimism1 | | .758 | | | | .752 | | | |
| Optimism2 | | .778 | | | | .781 | | | |
| Optimism3 | | .680 | | | | .683 | | | |
| Resil1 | | | .212 | | | | .282 | | |
| Resil2 | | | .658 | | | | .817 | | |
| Resil3 | | | .427 | | | | .385 | | |
| S-E1 | | | .629 | | | | | .633 | |
| S-E2 | | | .602 | | | | | .620 | |
| S-E3 | | | .720 | | | | | .759 | |
| Hope | | | | .867 | | | | | .789 |
| Optimism | | | | .595 | | | | | .582 |
| Resil. | | | | .791 | | | | | .863 |
| SelfEf | | | | | | | | | .821 |

*Note*: resil = resilience; S-E = self-efficacy; first-order f = first-order factor; 3f = 3 factor; 4f = 4 factor

problems and difficulties using the following three items: (10) *I am confident that I could deal efficiently with unexpected events*, (11) *I can solve most problems if I invest the necessary effort*, and (12) *I can remain calm when facing difficulties because I can rely on my coping*. Based on a comparison of items with the definition, we believe that the items fit the self-efficacy construct well.

Resilience is defined as *"when beset by problems and adversity, sustaining and bouncing back and even beyond to attain success.'* [4 p3]. In CPC-12, it is measured by the following three items: (7) *Sometimes I make myself do things whether I want to or not*, (8) *When I'm in a difficult situation, I can usually find my way out of it*, and (9) *It's okay if there are people who don't like me*. A comparison of the items with the definition shows that the content of items 7 and 9 is not in line with the definition of resilience, and that the items measure either an unrelated construct or, maybe, possible outcomes of high resilience. Item 8 is the closest to the definition of resilience, but it is also close to the definition of self-efficacy, as it includes a belief in one's own ability to cope with the problem. We believe that the content of resilience items is the reason that these items do not constitute a factor separate from self-efficacy. We conclude that while items intended to measure self-efficacy are in line with definition of self-efficacy, items intended to measure resilience represent definition of resilience only poorly. This led us to keep the items measuring self-efficacy and replace items measuring resilience.

To select new resilience items, we examined available resilience measures, especially the Brief Resilience Scale [52], Connor-Davidson Resilience Scale [53], and Resilience Scale for Adults [54], which received the best psychometric ratings in the review of resilience measurement scales performed by Windle et al. [55]. Inspired by the measures and guided by the above-mentioned definition of resilience, we created new items and subsequently chose three of them based on our assessment of how they represented the construct of resilience. The new items were the following:

- *I consider myself to be able to stand a lot, I am not easily discouraged by failure*,

- *After serious life difficulties, I tend to quickly bounce back*, and

- *I believe that coping with stress can strengthen me.*

We used these items as a replacement for the original resilience items in the new version, CPC-12R. In Study 3, we measured PsyCap using CPC-12R and tested the hypotheses that the data fit the theoretical model of PsyCap (i.e., four first-order factors: hope, self-efficacy, optimism, and resilience; one second-order factor: PsyCap) and that the model with four first-order factors is preferred over the model with three first-order factors (i.e., hope, optimism, and resilience+self-efficacy).

As part of the validation study, we supplemented the questionnaire with a fourth item ("I believe that coping with stress can strengthen me"), which also emerged from our assessment as a suitable item for measuring resilience. We added the fourth item as a spare item in case the statistical analysis showed another item to have insufficient psychometric characteristics.

## Materials and methods

**Psychological Capital.**    PsyCap was evaluated using a revised version of the original CPC Scale [29]. The revised version consisted of 9 original self-evaluating statements and 3 new resilience items rated on a 6-point Likert scale (ranging from 1 = *strongly disagree* to 6 = *strongly agree*). A fourth new item was introduced at the end of the questionnaire. This item was listed at the end so as to not affect how respondents would respond to individual CPC-12R items.

**Participants and procedure.**    The sample for Study 3 consisted of 333 respondents, of which 68% were women, 30% men, and 7 participants chose not to disclose gender. Average age of the participants was 35.74 years ($SD$ = 10.73). There were 2% who worked in primary production, 8% in manufacturing and industry, 47% in services and trade, 8% in state and local government, 22% in education, 4% in healthcare services, and 9% in non-profit sector. Participants were recruited by publishing a link to the survey in several online social media groups, and all participants were volunteers with no compensation. The survey was conducted in Czech. Participants were informed that they would give their consent by proceeding past the welcome page of the online survey. We did not seek ethics committee approval in this case as data were gathered anonymously and we did not expect any risks of harm for participants.

## Results

Table 8 contains descriptive statistics for all subscales and the compound PsyCap scale. The new resilience subscale was found to be internally consistent.

To assess the factorial structure of PsyCap in the CPC-12R, we conducted a confirmatory factorial analysis. All models were estimated with a MLR estimation in MPLUS 8 [38]. In a preliminary analysis, the higher-order model of PsyCap with 13 items (i.e., all four new resilience items), four first-order factors (i.e., hope, self-efficacy, optimism, and resilience), and

**Table 8. Descriptive statistics for summary scores of CPC-12R subscales and the compound scale.**

|               | M      | SD    | α    | ω    |
|---------------|--------|-------|------|------|
| Hope          | 13.545 | 2.600 | .761 | .771 |
| Optimism      | 14.451 | 3.101 | .793 | .806 |
| Resilience    | 12.949 | 2.918 | .701 | .729 |
| Self-efficacy | 14.156 | 2.437 | .891 | .895 |
| PsyCap        | 55.054 | 7.579 | .902 | 905  |

one second-order factor (i.e., PsyCap) indicated good fit according to Hu and Bentler [56] ($\chi^2$(61) = 141.919; CFI = .953; TLI = .939; RMSEA = .063; 90%CI$_{RMSEA}$ [.050, .077]). Among the 4 new items, the last spare item (13) had the lowest loading on the factor resilience ($\lambda$ = .534). Moreover, according to the residual correlation matrix, item 13 had the strongest residual correlations with other items. It correlated mainly with residuals of items from the optimism subscale.

Since we wanted to revise the psychological capital questionnaire to have 3 items in each subscale, just like in the original version, we excluded the spare item 13 from further analyses. The higher-order model of PsyCap with 12 items, four first-order factors and one second-order factor fit the data well and slightly better than the model with 13 items (see Table 9).

We compared this model with three alternative models. The model with one factor (i.e., all 12 items loaded on single factor PsyCap) did not fit the data well. The model that was preferred in Study 1 and Study 2 with three correlated factors (i.e., self-efficacy and resilience items loaded on a common factor) showed satisfactory but not good fit. The model with three first-order factors (i.e., hope, optimism, self-efficacy + resilience) and one second-order factor (i.e., PsyCap) did not converge. Only one alternative model, the model with four first order factors, fit the data well and showed similar fit to the hypothesized model with second-order factors. In further analyses we recommend using the second-order factor model because it had fewer parameters than the model with four correlated factors, and it also corresponded better to theory on psychological capital. The standardized factor loadings of all items were high ($\lambda \geq$ .669). Further, the first-order factors had high factor loadings on the second-order factor PsyCap (see Fig 1 for all factor loadings). The new resilience subscale correlated strongly with self-efficacy (latent factors: $r$ = .725, summary scores: $r$ = .626) and hope (latent factors: $r$ = .743, summary scores: $r$ = .562), and moderately with optimism (latent factors: $r$ = .598, summary scores: $r$ = .365).

## Discussion

Showing that the content of resilience items in the original CPC-12 scale is not in line with the definition of resilience as a facet of psychological capital, we replaced these items with new items that were formulated to better reflect the definition [3]. According to the CFA, the model with three first-order factors (i.e., hope, optimism and resilience/self-efficacy) and one second-order factor, which was preferred in studies using CPC-12, did not converge. On the other hand, the analyses showed a good fit of data collected by new CPC-12R to the theoretical model of PsyCap with four first-order factors and one second-order factor. Unlike the original resilience items, the new items loaded strongly on the common factor of resilience, supporting the factorial validity of CPC-12R.

We also provided evidence about reliability of the compound PsyCap scale and particular subscales, showing their high internal consistency.

## General discussion

This article describes limitations of the original CPC-12 scale published by Lorenz et al. [29]. The secondary analyses on both original German samples in Study 2 and a new analysis on a Czech sample in Study 1 showed that the theoretical hierarchical four-factor model of psychological capital did not describe the data collected by CPC-12 better than a more parsimonious hierarchical model with three first-order factors. We compared the content of the CPC-12 items with the definition of related constructs and found that the original resilience items did not reflect the definition of resilience well. We formulated new items according to the definition of resilience [3], and we suggested their use instead of original resilience items in the

**Table 9. CFA: Comparison of alternative models.**

| Model | Description | $\chi^2$ | df | P | CFI | TLI | RMSEA [90%CI] | SRMR |
|-------|-------------|---------|----|----|-----|-----|----------------|------|
| Baseline | Uncorrelated items | 1685.849 | 66 | < .001 | < .001 | < .001 | .271[.260, .283] | .391 |
| 1 factor | 12 items on PsyCap | 499.723 | 54 | < .001 | .725 | .664 | .157 [.145, .170] | .090 |
| 3 factors | S-E + resil. together | 176.514 | 51 | < .001 | .923 | .900 | .086 [.072, .100] | .052 |
| 4 factors | 4 correlated factors | 104.679 | 48 | < .001 | .965 | .952 | .060 [.044, .075] | .040 |
| 2nd order 4f | 4 first-order f. + PsyCap | 110.394 | 50 | < .001 | .963 | .951 | .060 [.045, .075] | .043 |

*Note*: resil = resilience; first-order f = first-order factor; 3f = 3 factor; 4f = 4 factor

revised CPC-12R. In Study 3, we provided evidence about the factorial validity of CPC-12R and about the internal consistency of the compounded PsyCap scale and all four subscales. The revised version of the instrument demonstrated better psychometric characteristics than the original CPC-12 [29]. In CPC-12R, the difference between subscales measuring self-effi-cacy and resilience was strengthened; consequently, the factorial structure of CPC-12R fit the structure supported by literature [1] better than CPC-12 (ibid). We recommend using the revised CPC-12R instead of the original scale when measuring PsyCap.

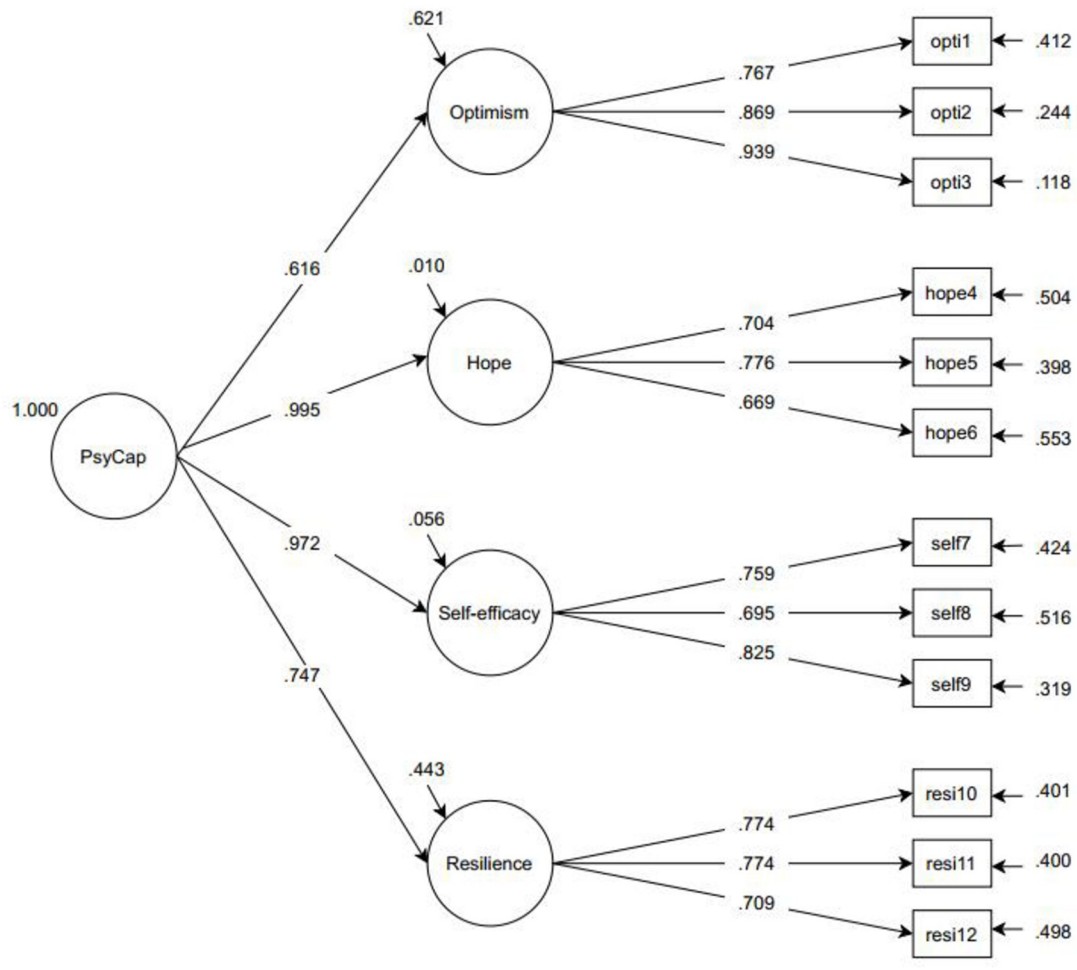

**Fig 1. Measurement model of the CPC-12R.**

### Limitations and future research

Recognition of the limits of generalizability is important. Convenience sampling was used in all three studies, and the demographics of participants differed in the three population distributions. While the psychometric limitations of the CPC-12 based on analysis of three different datasets using two different language versions seem convincing, the psychometric characteristics of the CPC-12R were documented on only one sample that was more educated, younger, and with a higher proportion of women compared to the general population. As such, our findings should be understood as a contribution to the process of development of a free-to-use PsyCap measure with universal claim, and future studies are needed to complement our findings. Nevertheless, we have no reason to assume that convenience sampling influenced the factorial structure of the measure. Comparing the CPC-12R with samples from German studies [29], we can see that the three scales that are common to CPC-12 and CPC-12R have comparable psychometric characteristics for all three samples. Moreover, the utility of findings based on convenience samples is supported by further evidence from earlier studies [57, 58].

Lorenz et al. [29] provided evidence about the concurrent validity of CPC-12 by comparing PsyCap measured by CPC-12 with PsyCap measured by PCQ. Furthermore, they provided evidence about the construct validity of CPC-12 using correlation with other theoretically related constructs. We assume that similar results should be obtained using CPC-12R as we replaced only three items and as the new resilience factor was correlated to other factors in accordance with the theory. However, future studies are recommended to provide support for psychometric qualities of the revised scale, especially in terms of concurrent and construct validity.

To broaden possible use of CPC-12R, translation and verification of the validity of the instrument in different languages is required. Furthermore, validation on a sample representative of the working population and development of standards would facilitate the individual use of the questionnaire in organizations.

## Conclusion

The important impact of psychological capital on job attitudes and behaviors has been widely documented, yet follow-up research and its use in practice has so far been complicated by difficult-to-access measurement methods. In this study, we provide a revised version of a freely available compound measure for PsyCap, with the general claim of being applicable not only in the work environment, but also in other domains of life. Future research validating this method in different language and cultural contexts is encouraged.

## Supporting information

**S1 Dataset. Dataset study 1 ($N$ = 282).**
(SAV)

**S2 Dataset. Dataset study 2 –first sample ($N$ = 321).**
(SAV)

**S3 Dataset. Dataset study 2 –second sample ($N$ = 202).**
(SAV)

**S4 Dataset. Dataset study 3 ($N$ = 333).**
(SAV)

**S1 Appendix. CPC-12R in Czech and English.**
(DOCX)

## Acknowledgments

Thanks to Vojtěch Hercík for help with data collection.

## Author Contributions

**Conceptualization:** Ludmila Dudasova, Jakub Prochazka, Martin Vaculik.

**Data curation:** Ludmila Dudasova, Timo Lorenz.

**Formal analysis:** Ludmila Dudasova, Jakub Prochazka, Timo Lorenz.

**Funding acquisition:** Ludmila Dudasova, Jakub Prochazka, Martin Vaculik.

**Project administration:** Martin Vaculik.

**Writing – original draft:** Ludmila Dudasova, Jakub Prochazka.

**Writing – review & editing:** Ludmila Dudasova, Jakub Prochazka, Martin Vaculik, Timo Lorenz.

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
