## [Decision Letter · Decision Letter 0]

13 Oct 2020

PONE-D-20-21923

Measuring Positive Psychological Capital: Revision of the CPC-12

PLOS ONE

Dear Dr.
Kašpárková,

Thank you for submitting your manuscript to PLOS ONE. After careful consideration, we
feel that it has merit but does not fully meet PLOS ONE’s publication criteria as it
currently stands. Therefore, we invite you to submit a revised version of the
manuscript that addresses the points raised during the review process.

Please submit your revised manuscript by Nov 26 2020 11:59PM. If you will need more
time than this to complete your revisions, please reply to this message or contact
the journal office at plosone@plos.org. When
you're ready to submit your revision, log on to https://www.editorialmanager.com/pone/ and select the 'Submissions
Needing Revision' folder to locate your manuscript file.

If you would like to make changes to your financial disclosure, please include your
updated statement in your cover letter. Guidelines for resubmitting your figure
files are available below the reviewer comments at the end of this letter.

We look forward to receiving your revised manuscript.

Kind regards,

Meng-Cheng Wang

Academic Editor

PLOS ONE

Journal Requirements:

3. Please improve statistical reporting and refer to p-values as "p<.001" instead
of "p=.000". Our statistical reporting guidelines are available at https://journals.plos.org/plosone/s/submission-guidelines#loc-statistical-reporting.

Reviewers' comments:

Reviewer's Responses to Questions

**Comments to the Author**

1. Is the manuscript technically sound, and do the data support the conclusions?

Reviewer #1: Yes

Reviewer #2: Partly

2. Has the statistical analysis been performed
appropriately and rigorously? 

Reviewer #1: Yes

Reviewer #2: N/A

3. Have the authors made all data underlying the
findings in their manuscript fully available?

Reviewer #1: Yes

Reviewer #2: Yes

4. Is the manuscript presented in an intelligible
fashion and written in standard English?

Reviewer #1: Yes

Reviewer #2: Yes

5. Review Comments to the Author

Reviewer #1: For both exploratory factor analysis (EFA) and confirmative factor
analysis (EFA)，the matric of factor loadings is very useful information besides
fitness indices of models，such that I suggest the authors should present specific
factor loadings or factor measurement model of CFA to help reader understanding.

Reviewer #2: The study " Measuring Positive Psychological Capital: Revision of the
CPC-12" is clear and well-written. I found the purpose of the study very practical.
Yet, I have more concerns about the hypotheses, the analysis method used and the
theoretical background of the paper. Such major concerns should be further taken
into consideration in this paper.

1. According to the title and the part of Materials and Methods, the target construct
was positive psychological capital, but in the introduction part, only the construct
of psychological capital was mentioned. The relation of positive psychological
capital and psychological capital should be provided or the expression should be
consistent.

2. The most significant problem I found was that the hypotheses of the current study
was unclear, which made the study design data-driven. As I understand, the central
purpose was to develop a Czech version of Compound Psychological Capital Scale. In
the first study, the results showed that the construct validity was not confirmed.
To this end, the authors should consider modifying the items or discussing other
constructs of psychological capital. I didn’t find much need to do the secondary
analysis of the original data of Lorenz et al, because I didn’t see the hypotheses
to guide the analysis.

3. The other issue in the current study was that the theoretical background was
limited. In the introduction part, except for PCQ and CPC-12, not much information
was given on the measurement of psychological capital. How the dimensions of
psychological capital were defined by other studies? Is the second-order four-factor
construct supported in other studies? How the self-efficacy and resilience related?
The authors should provide more evidence to support your study. More important,
positive and negative evidence about the validity of CPC-12 should be provided,
considering the current study was based on the adaptation of CPC-12. Those evidence
could be the foundation of subsequent analysis of results.

4. I have some confusion about the analysis method used. In three studies, MLR was
used as in the study of Lorenz et al. However, estimation was determined mainly by
the data pattern. How it’s verified that MLR was the most appropriate estimator?
Also, it’s unclear why Varimax rotation was used, instead of oblique rotation
method.

5. In the results part, more important information should be added, including the
descriptive statistics, reliability coefficients, and the full factor loadings in
each study. The results mainly focused on the model fit, while it’s also essential
to analyze how items were distributed to different factors. For example, in study 1,
it said that “Although there were a number of cross-loadings , items from the hope
subscale tended to load on one common factor, ……”, how the items were distributed
should also be analyzed except for the model fit results. Besides, the 3-factor
model was not significantly better fitted than 4-factor model. It’s not adequate to
conclude that “three-factor model of PsyCap fit the data better”.

6. In the discussion part of study 1, the authors provided three assumptions to
explain the results. Which assumption did the authors support, according to the
theoretical evidence and the results? How did the previous studies find the
relations between self-efficacy and resilience? The discussion lacked evidence to
support the opinions.

7. According to the analysis of study 1, in Lorenz’s study, the model included “12
items, four first-order factors (i.e., hope, self-efficacy, optimism, and
resilience), and one second-order factor”. Why the models in study 2 were all
one-order models in EFA? (“the analyses on both datasets identified one, two, and
three-factor solutions”). What is the purpose of EFA? Besides, according to the
results in Table 4, the difference between 3-factor and 4-factor model was not
significant.

8. In study 3, why items on resilience were modify instead of self-efficacy? To adapt
one subscale, only four new items were prepared. It was not adequate. Additionally,
How the removed items were determined? According to the article, “To limit
possibility of bias caused by the order of items, we placed the fourth reserve item
measuring resilience at the end of the questionnaire.” How did the author know that
the fourth item will be removed in advance?

9. In discussion of study 3, it said that “There was a difference in the ratio of
items measuring individual components compared to PCQ-12”. However, how items were
modified was not introduced in advance, which caused much confusion.

10. For a study to develop scales, reliability evidence and more validity evidence
should be provided. Though it was mentioned in the limitations, it remained a fatal
flaw.

6. PLOS authors have the option to publish the peer
review history of their article (what does this mean?). If published, this will
include your full peer review and any attached files.

If you choose “no”, your identity will remain anonymous but your review may still be
made public.

**Do you want your identity to be public for this peer review?** For
information about this choice, including consent withdrawal, please see our
Privacy Policy.

Reviewer #1: No

Reviewer #2: No

---

## [Author Response · Author response to Decision Letter 0]

1 Dec 2020

Responses to reviewers’ comments: 

Reviewer 1: 

Response: We want to thank Reviewer 1 for their positive evaluation of our study and
for the following comment that helped us in improving our manuscript.

For both exploratory factor analysis (EFA) and confirmative factor analysis (CFA),
the matric of factor loadings is very useful information besides fitness indices of
models，such that I suggest the authors should present specific factor loadings or
factor measurement model of CFA to help reader understanding.

Response: Thank you for your remark. We share the conviction that factor loadings
represent a very useful information alongside fitness indices of the models. 

Responding to feedback from the second reviewer, we decided to remove EFA from the
manuscript. 

The purpose of the EFA was to show that the data cannot be better explained by any
other model than one of the models derived from the theory. However, we agree with
the second reviewer that EFA is not necessary to meet the objectives of the study,
and that CFAs provide all the information needed to test our hypotheses.
Nevertheless, if you find it relevant, we are prepared to insert it back, and to add
a table with factor loadings. 

In relation to CFA, we have added new tables with factor loadings in Study 1 (Table
2, p. 9) and Study 2 (Table 6, page 15; Table 7, p. 16). In Study 2, specific factor
loadings have been presented for models with three and four first-order factors. In
Study 3, the factor loadings are available in the Figure.

Reviewer #2: 

Response: We want to thank Reviewer 2 for reading our manuscript carefully and for
providing their valuable comments. They helped us in making the aim and
contributions of our manuscript clearer and in adding the missing information
important for understanding the procedures and analyses.

The study " Measuring Positive Psychological Capital: Revision of the CPC-12" is
clear and well-written. I found the purpose of the study very practical. Yet, I have
more concerns about the hypotheses, the analysis method used and the theoretical
background of the paper. Such major concerns should be further taken into
consideration in this paper.

1. According to the title and the part of Materials and Methods, the target construct
was positive psychological capital, but in the introduction part, only the construct
of psychological capital was mentioned. The relation of positive psychological
capital and psychological capital should be provided or the expression should be
consistent.

Response: Thank you carefully reading our text. We omitted the word “positive” in the
first version of the manuscript. Thanks to your comment, we questioned whether the
word indeed has a function and concluded that we can continue using the term
“psychological capital” (as used in the original article introducing CPC-12 scale).
Therefore, we replaced the terms “positive psychological capital” with
“psychological capital.” 

2. The most significant problem I found was that the hypotheses of the current study
was unclear, which made the study design data-driven. As I understand, the central
purpose was to develop a Czech version of Compound Psychological Capital Scale. In
the first study, the results showed that the construct validity was not confirmed.
To this end, the authors should consider modifying the items or discussing other
constructs of psychological capital. I didn’t find much need to do the secondary
analysis of the original data of Lorenz et al, because I didn’t see the hypotheses
to guide the analysis.

Response: Thank you for your suggestion. We recognize the need to clarify the
rationale of our research. Our original goal was to adapt a Czech version of the
questionnaire. With this intention, we designed and executed the first study.
However, based on the analysis of the newly collected data, the goal of the study
changed and now the aim of the manuscript is to draw attention to the significant
limitations of the published questionnaire CPC-12 and to propose a revised version.
Please see the second paragraph on p. 6, where we explain the logic of our
study.

None of the studies were data-driven. In the first study, we began with the
hypothesis that the data obtained by the Czech adaptation of CPC-12 would correspond
to the theoretical four-factor model of positive psychological capital (p. 6, third
paragraph). However, we did not find support for this hypothesis. Consequently, we
analyzed the data and searched for an explanation for this result. We found out that
items from the resilience and self-efficacy subscales tended to load on a single
common factor. Initially, we focused on the content of the Czech items, believing
that inaccurate translation had led to such results. However, we concluded that the
multi-stage translation was performed well, and the content of the Czech items
corresponded to the content of the original items. 

Therefore, we decided to inspect the data from the original German study and
hypothesized that we would find the same problem with the factor structure (p. 12,
lines 265-269). In Study 2, this hypothesis was proven correct; we found the same
problem which provided important evidence about limitation of the original scale. We
consider this to be a very important finding as it shows that the article previously
published in PLOS ONE does not contain precise conclusions, and that CPC-12 does not
measure positive psychological capital as a construct with four first-order factors.
The aim was now to inspect the possible cause of the findings. There were three
possible causes to investigate (p. 10): translation error, an error in PsyCap theory
and limitations of the original CPC-12. Upon further analysis, we found that the
items intended to measure resilience do not match the content of the resilience
construct. This conclusion (low factor loadings of resilience items, low reliability
of the resilience subscale) (p. 17, second paragraph) led to study 3.

Consequently, we proposed new items for measuring resilience so that they better
match the content of the construct and that the CPC-12R scale measures all 4
components of positive psychological capital. In the third study, we tested whether
the data obtained using CPC-12R fit the theoretical 4+1 factor model and also if the
4+1 model of PsyCap explained the data better than the more parsimonious model with
3+1 factors which was preferred when using the original scale (p. 17, third
paragraph). We found support for these hypotheses and provided evidence on factorial
validity and reliability of the revised scale. We also suggested directions for
further validation of the revised questionnaire. 

We believe that all three studies are important for our article. The first study
explains our motivation to revise the scale and provide independent evidence about
its psychometric limitations. The second study draws attention to the problem in the
interpretation of data in the study previously published by Lorenz et al. (2016) and
to the shortcomings of the published scale (CPC-12). The third study proposes a new
version of the scale (CPC-12R) which deals with the previously mentioned limitations
and provides initial evidence on factorial validity and reliability of the revised
scale.

We agree that the hypotheses were not evident from the text of the manuscript. Thanks
to your feedback, we have now explicitly stated the hypotheses for all three
studies.

3. The other issue in the current study was that the theoretical background was
limited. In the introduction part, except for PCQ and CPC-12, not much information
was given on the measurement of psychological capital. How the dimensions of
psychological capital were defined by other studies? Is the second-order four-factor
construct supported in other studies? How the self-efficacy and resilience related?
The authors should provide more evidence to support your study. More important,
positive and negative evidence about the validity of CPC-12 should be provided,
considering the current study was based on the adaptation of CPC-12. Those evidence
could be the foundation of subsequent analysis of results.

Response: Thank you for your suggestion to implement more details about PsyCap
measurement. Please see pp. 4-5, where we provide additional information. Other
studies define the dimensions of PsyCap comparably to how we defined it (see, e. g.
Luthans, Avey and Patera, 2008, listed as citation number 5 in our manuscript) and
the second-order four-factor construct is supported by other studies. Reader can
find this information on p. 2, lines 42-43 (“The existence of a higher order core
construct, PsyCap, has both conceptual [1] and empirical [5] support.”).

To help reader understand the relationship between resilience and self-efficacy, we
start by providing detailed definitions of the constructs. We note that these
constructs are close, hence we describe the shared content as well as the difference
between them. According to research that has examined the relationship between these
variables, the relationship between them is significant, but not so strong that we
doubt that they are two different constructs. Please see pp. 10-11 (lines 223-234). 

When we found the overlap between them, we examined carefully both the items
measuring resilience and self-efficacy. We concluded that while items intended to
measure self-efficacy are in line with the definition of self-efficacy, items
intended to measure resilience do not reflect the definition of resilience. This led
us to keep the items measuring self-efficacy and replace the items measuring
resilience. Thanks to your suggestion, we describe it now in the manuscript more
clearly (see pp. 17-18, lines 353-374).

As for the positive and negative evidence about the validity of CPC-12, the number of
published studies examining the psychometric properties of the scale remain very
limited as the scale was published recently. However, in response to your comment,
we addressed this on p. 5 (see changes on lines 111-113).

4. I have some confusion about the analysis method used. In three studies, MLR was
used as in the study of Lorenz et al. However, estimation was determined mainly by
the data pattern. How it’s verified that MLR was the most appropriate estimator?
Also, it’s unclear why Varimax rotation was used, instead of oblique rotation
method.

Response: Because items had a sufficiently large response scale to be approximated as
an interval variable, we used maximum likelihood estimation instead of estimator
that assumes ordinal nature of the data. Due to the rather small sample size,
results using the maximum likelihood robust estimation are less likely to be biased
than results using WLSMV or similar estimator ( for details see Rhemtulla et al.,
2012, Sass et al., 2014 and Li, 2016 ). We believe that the use of a robust maximum
likelihood estimator for CFA of scales with a five-point response scale is so
widespread that it does not need to be specifically justified. It is also noteworthy
that the original study did not mention the estimator and it is not common to
justify the estimator on PLOS ONE manuscripts. However, if the reviewer wishes, we
can add the above-mentioned explanation to the manuscript.

We used the predefined Varimax rotation for the EFA. We agree that the oblique
rotation would be more appropriate given that the model assumes highly correlated
factors. We recalculated the results using GEOMIN (OBLIQUE) rotation and the outputs
are comparable to the original outputs with Varimax rotation. However, according to
one of your comments (mentioned ahead) we have removed the EFA from the manuscript
as the analysis did not contribute significantly to the aim of the manuscript (see
below), but we are ready to insert it again if so advised.

References: 

Li, C. H. (2016). Confirmatory factor analysis with ordinal data: Comparing robust
maximum likelihood and diagonally weighted least squares. Behavior research methods,
48(3), 936-949. https://doi.org/10.3758/s13428-015-0619-7

Rhemtulla, M., Brosseau-Liard, P. É., & Savalei, V. (2012). When can categorical
variables be treated as continuous? A comparison of robust continuous and
categorical SEM estimation methods under suboptimal conditions. Psychological
methods, 17(3), 354-373. https://doi.org/10.1037/a0029315

Sass, D. A., Schmitt, T. A., & Marsh, H. W. (2014). Evaluating model fit with
ordered categorical data within a measurement invariance framework: A comparison of
estimators. Structural Equation Modeling: A Multidisciplinary Journal, 21(2),
167-180. 

https://doi.org/10.1080/10705511.2014.882658

5. In the results part, more important information should be added, including the
descriptive statistics, reliability coefficients, and the full factor loadings in
each study. The results mainly focused on the model fit, while it’s also essential
to analyze how items were distributed to different factors. For example, in study 1,
it said that “Although there were a number of cross-loadings , items from the hope
subscale tended to load on one common factor, ……”, how the items were distributed
should also be analyzed except for the model fit results. Besides, the 3-factor
model was not significantly better fitted than 4-factor model. It’s not adequate to
conclude that “three-factor model of PsyCap fit the data better”.

Response: We have added tables containing descriptive statistics, reliability
coefficients, and factor loadings for all studies to the manuscript. 

Thank you for pointing out the misinterpretation of the comparison of the two models.
We have corrected the text (see changes on p. 8).

6. In the discussion part of study 1, the authors provided three assumptions to
explain the results. Which assumption did the authors support, according to the
theoretical evidence and the results? How did the previous studies find the
relations between self-efficacy and resilience? The discussion lacked evidence to
support the opinions.

Response: Thank you for your valuable comment. We have made significant changes in
the discussion of study 1 to help reader understand our decisions (please see
changes on pp. 9-12). 

The relationship between resilience and self-efficacy had already been investigated.
We now added that the previous studies found these two concepts to be moderately to
strongly related. However, the studies also showed that resilience and self-efficacy
are correlated but different and not linearly dependent constructs, and that factor
analysis (using PCQ) is able to distinguish among them. Our finding that resilience
and self-efficacy cannot be distinguished are not in line with previous studies. We
hope that it is now more evident from the text of the manuscript.

7. According to the analysis of study 1, in Lorenz’s study, the model included “12
items, four first-order factors (i.e., hope, self-efficacy, optimism, and
resilience), and one second-order factor”. Why the models in study 2 were all
one-order models in EFA? (“the analyses on both datasets identified one, two, and
three-factor solutions”). What is the purpose of EFA? Besides, according to the
results in Table 4, the difference between 3-factor and 4-factor model was not
significant.

Response: Thank you for raising this issue. It is not possible to perform EFA with a
second-order factor. For second-order factor models, multiple EFAs are made, one for
each second-order factor. Because all four components of PPC are saturated with the
same second-order factor, we consider our approach to be correct.

The purpose of the EFA was to show that the data cannot be better explained by any
other model than one of the models derived from the theory. However, we agree that
EFA is not necessary to meet the objectives of the study and it is not appropriate
for the study that does not want to be exploratory and data driven. CFAs provide all
the information needed to test our hypotheses. Hence, we have removed the EFA from
the manuscript. However, we are ready to insert it again if you so recommend.

8. In study 3, why items on resilience were modify instead of self-efficacy? To adapt
one subscale, only four new items were prepared. It was not adequate. Additionally,
How the removed items were determined? According to the article, “To limit
possibility of bias caused by the order of items, we placed the fourth reserve item
measuring resilience at the end of the questionnaire.” How did the author know that
the fourth item will be removed in advance?

Response: We replaced the resilience items because their original content did not
accurately reflect the content of the resilience construct. This is not the case
with the self-efficacy subscale; items in it adequately reflect the self-efficacy
construct. We emphasized this explanation in the text of the study to make it clear
(pp. 17-18). We now also show more precisely the empirical evidence of the low
content validity of the original resilience subscale.

While we created several new items, we chose the three items which we thought
described the content of the resilience construct well. These items covered the
whole construct of resilience (as it is defined as a part of the PsyCap definition).
We chose the fourth spare item and added it in the end of the questionnaire in case
the analysis revealed a problem in one of the first three proposed items. However,
as the first three items worked as expected, we left the spare item from further
analyses. We wanted the revised version to have the same number of items as the
original version, and for all subscales to have the same number of items.

We agree that including only four items in the data collection could represent a risk
if more than one item showed to be problematic. However, this has not happened.

9. In discussion of study 3, it said that “There was a difference in the ratio of
items measuring individual components compared to PCQ-12”. However, how items were
modified was not introduced in advance, which caused much confusion.

Response: Thank you for your valuable comment. We agree that this part of the text
was confusing. We did not change the number of items measuring each component. We
only wanted to inform the readers that CPC-12 measures PsyCap by 3+3+3+3 items, and
therefore the effects of all components are balanced, which is a benefit compared to
PCQ-12 which is composed of 3+4+3+2 items. Following your comment, we removed this
information from the discussion of study 3 and inserted it in the introduction of
study 1, where we introduce and compare existing scales (p. 5, first paragraph).

10. For a study to develop scales, reliability evidence and more validity evidence
should be provided. Though it was mentioned in the limitations, it remained a fatal
flaw.

Response: Thank you for raising this concern. The main contribution of our manuscript
is not the new scale, but the fact, that we found a psychometric limitation of the
original scale that was published in PLOS ONE. However, we did not just want to find
limitations and leave it at that. We also wanted to show that the problem can be
solved by replacing items that were used for measuring resilience. The contribution
of Study 3 is not only in providing the revised scale, but also in showing that we
found the correct cause of the problem that we identified in Study 1 and Study 2 and
that the problem can be solved.

If the aim was to develop a new scale, we would arrange Study 1 as a study focused on
developing and choosing the best items and we would add further studies to provide
evidence about the validity of the new scale. However, our aim was different. We
presented two studies that highlighted the problem in a published and widely cited
scale and we arranged the third study to suggest the solution. We agree that further
studies are needed to provide more evidence on the validity of the revised scale,
and we state it in the general discussion (see p. 24, lines 507-510).

to Reviewers.docx
---

## [Decision Letter · Decision Letter 1]

13 Jan 2021

PONE-D-20-21923R1

Measuring Psychological Capital: Revision of the Compound Psychological Capital Scale
(CPC-12)

PLOS ONE

Dear Dr. Kašpárková,

Thank you for submitting your manuscript to PLOS ONE. After careful consideration, we
feel that it has merit but does not fully meet PLOS ONE’s publication criteria as it
currently stands. Therefore, we invite you to submit a revised version of the
manuscript that addresses the points raised during the review process.

Please submit your revised manuscript by Feb 27 2021 11:59PM. If you will need more
time than this to complete your revisions, please reply to this message or contact
the journal office at plosone@plos.org. When
you're ready to submit your revision, log on to https://www.editorialmanager.com/pone/ and select the 'Submissions
Needing Revision' folder to locate your manuscript file.

If you would like to make changes to your financial disclosure, please include your
updated statement in your cover letter. Guidelines for resubmitting your figure
files are available below the reviewer comments at the end of this letter.

We look forward to receiving your revised manuscript.

Kind regards,

Meng-Cheng Wang

Academic Editor

PLOS ONE

Reviewers' comments:

Reviewer's Responses to Questions

**Comments to the Author**

1. If the authors have adequately addressed your comments raised in a previous round
of review and you feel that this manuscript is now acceptable for publication, you
may indicate that here to bypass the “Comments to the Author” section, enter your
conflict of interest statement in the “Confidential to Editor” section, and submit
your "Accept" recommendation.

Reviewer #1: All comments have been addressed

Reviewer #2: All comments have been addressed

2. Is the manuscript technically sound, and do the data
support the conclusions?

Reviewer #1: Yes

Reviewer #2: Yes

3. Has the statistical analysis been performed
appropriately and rigorously? 

Reviewer #1: Yes

Reviewer #2: N/A

4. Have the authors made all data underlying the
findings in their manuscript fully available?

Reviewer #1: Yes

Reviewer #2: Yes

5. Is the manuscript presented in an intelligible
fashion and written in standard English?

Reviewer #1: Yes

Reviewer #2: Yes

6. Review Comments to the Author

Reviewer #1: I think the authors have addressed comments from the reviewers and the
draft had been revised adequately.

Reviewer #2: 1. In the descriptive statistics, “mean scores” in Table 1 and Table 8
were sum scores averagely, which were different from Table 3. It’s better to keep
them consistent.

2. It should be “higher CFI”, not “higher CFA” (p. 14, line 297).

3. The results in study 2 showed better fit indices in the 3-factor model in the
first German dataset, and better fit indices in the 4-factor model in the second
German dataset. It was confusing that why both results supported your hypotheses,
considering they were not consistent. If the small difference in the fit indices
(ΔCFI < .01, ΔRMSEA < .005) was not the evidence of “significant improvement
in the model's fit”, as you stated in the explanation of results of the second
dataset, then why the 3-factor model was better than 4-factor model in the first
dataset?

7. PLOS authors have the option to publish the peer
review history of their article (what does this mean?). If published, this will
include your full peer review and any attached files.

If you choose “no”, your identity will remain anonymous but your review may still be
made public.

**Do you want your identity to be public for this peer review?** For
information about this choice, including consent withdrawal, please see our
Privacy Policy.

Reviewer #1: No

Reviewer #2: No

---

## [Author Response · Author response to Decision Letter 1]

19 Jan 2021

Responses to reviewers’ comments: 

Reviewer 2: 

1. In the descriptive statistics, “mean scores” in Table 1 and Table 8 were sum
scores averagely, which were different from Table 3. It’s better to keep them
consistent.

Response: Thank you for this remark. We changed the values in Table 3 to display the
mean of summary scores.

2. It should be “higher CFI”, not “higher CFA” (p. 14, line 297).

Response: Thank you for careful reading and for finding this mistake. We fixed
it.

3. The results in study 2 showed better fit indices in the 3-factor model in the
first German dataset, and better fit indices in the 4-factor model in the second
German dataset. It was confusing that why both results supported your hypotheses,
considering they were not consistent. If the small difference in the fit indices
(ΔCFI < .01, ΔRMSEA < .005) was not the evidence of “significant improvement
in the model's fit”, as you stated in the explanation of results of the second
dataset, then why the 3-factor model was better than 4-factor model in the first
dataset? 

Response: The model with three factors is more parsimonious model which is nested in
model with four factors. The less parsimonious model should have significantly
better fit to be preferred over the more parsimonious model. As both models have
very similar fit indexes (ΔCFI < .01), the four-factor model should be not
preferred over the three-factor model (as described in Cheung GW, Rensvold RB,
2002). Therefore, the model with three factors is preferred which provides support
for our hypothesis. We added this information to the manuscript (see p. 14).

Cheung GW, Rensvold RB. Evaluating goodness-of-fit indexes for testing measurement
invariance. Struct Equ Modeling. 2002 Apr 1;9(2):233-55. https://doi.org/10.1207/S15328007SEM0902_5

---

## [Editor Report · Decision Letter 2]

2 Feb 2021

Measuring Psychological Capital: Revision of the Compound Psychological Capital Scale
(CPC-12)

PONE-D-20-21923R2

Dear Dr.
Dudášová

We’re pleased to inform you that your manuscript has been judged scientifically
suitable for publication and will be formally accepted for publication once it meets
all outstanding technical requirements.

Kind regards,

Meng-Cheng Wang

Academic Editor

PLOS ONE
---

## [Editor Report · Acceptance letter]

5 Feb 2021

PONE-D-20-21923R2 

Measuring Psychological Capital: Revision of the Compound Psychological Capital Scale
(CPC-12) 

Dear Dr. Dudasova:

I'm pleased to inform you that your manuscript has been deemed suitable for
publication in PLOS ONE. Congratulations! Your manuscript is now with our production
department. 

Kind regards, 

on behalf of

Dr. Meng-Cheng Wang 

Academic Editor

PLOS ONE